# OpenReview forum: "WAGLE: Strategic Weight Attribution for Effective and Modular Unlearning in Large Language Models"
_NeurIPS.cc/2024/Conference — NeurIPS 2024 poster_

### Official Review · Reviewer_QvH1 · 2024-07-11

**Soundness:** 4
**Presentation:** 3
**Contribution:** 3
**Rating:** 7
**Confidence:** 4

**Summary:**

This paper introduces WAGLE (Weight Attribution-Guided LLM unLEarning), a novel method for enhancing the effectiveness of large language model (LLM) unlearning. The key contributions in my view are:

1) A bi-level optimization framework for weight attribution in LLM unlearning.
2) A closed-form solution for calculating weight attribution scores.
3) Integration of weight attribution into existing LLM unlearning methods. The authors evaluate WAGLE across multiple unlearning tasks and benchmarks, demonstrating improved unlearning efficacy while maintaining model utility compared to baselines.

**Strengths:**

Strengths:
- Novel approach to LLM unlearning using weight attribution
- Theoretical grounding in bi-level optimization
- Comprehensive evaluation across multiple unlearning tasks and benchmarks
- Compatibility with existing unlearning methods (GradDiff, NPO, PO)
- Insights into which model components are most influential for unlearning

**Weaknesses:**

Weaknesses:
- Limited theoretical justification for why weight attribution improves unlearning
- Computational complexity of the method not thoroughly discussed
- Lack of comparison to some recent unlearning methods
- Limited exploration of potential negative effects or failure cases
- Hyperparameter sensitivity (e.g., choice of γ) could be explored more thoroughly

**Questions:**

Questions:
1. How does the computational complexity of WAGLE compare to baseline unlearning methods?
2. Have you explored the potential for negative transfer or catastrophic forgetting when using WAGLE?
3. How sensitive is the method to the choice of hyperparameters, particularly the Hessian diagonal estimate γ?
4. Could you provide more intuition on why certain model components (e.g., self-attention output and value) are more important for unlearning?
5. Have you considered applying WAGLE to other model architectures beyond LLaMA and Zephyr?
6. How does WAGLE perform on even larger language models (e.g., 13B or 70B parameter models)?
7. Could the weight attribution method be used for other purposes beyond unlearning, such as model compression or transfer learning?
8. Have you explored the potential for combining WAGLE with other unlearning techniques, such as data augmentation or adversarial training?

**Limitations:**

Limitations:
The authors have addressed some limitations, but the paper could improve by:

1. Addressed:

- Sensitivity to the Hessian diagonal hyperparameter γ
- Performance across different unlearning tasks and methods
- Comparison to several baseline methods

2. Could be better addressed:
- Computational complexity and scalability to larger models
- Potential negative effects or failure cases
- Broader applicability beyond the tested models and tasks

3. Missing:
- Discussion of potential biases introduced by the method
- Exploration of privacy implications
- Analysis of the method's robustness to adversarial attacks
- Environmental impact of the additional computational requirements

Suggestions for improvement:
1. Include a dedicated section on limitations and future work
2. Discuss broader ethical implications and potential misuse of the technology
3. Consider looking into relearning papers which I think are related but missing in this draft e.g. https://arxiv.org/pdf/2401.01814

---

> ### Author Rebuttal · Authors · 2024-08-07
>
> Thank you for the constructive feedback. Please find our response to each weakness (W), question (Q), and limitation (L) below. References (in the format of [Rx]) can be found in the general response.
>
> **Response to W1:** Our work, while lacking formal theoretical guarantees on how weight attribution enhances unlearning, is rooted in a rigorous bi-level problem formulation (Eq. (2)), leading to our final score derivation (Eq. (8)). This has deepened our understanding of LLM weight impacts on unlearning, especially in maintaining utility. For motivation and benefits of weight attribution in improving the unlearning efficacy-utility tradeoff, please see **GR1**. While a stronger theoretical guarantee could improve our work, this limitation does not undermine our significant technical contributions. We will discuss this point in the Limitations section.
>
> **Response to W2&Q1:** Please refer to **GR2** in the general response.
>
> **Response to W3:** We appreciate your comment on including recent unlearning methods. At submission, NPO (published April 8, 2024) was the most advanced, leading in the TOFU and the very recent MUSE benchmarks (after submission) [R10]. In our revision, we will survey and include any new SOTA methods published post-submission.
>
>
> **Response to W4:**  In response to your suggestion, we've considered potential drawbacks of our method, particularly the reliance of WAGLE on the Hessian diagonal estimate, $\gamma$, in Eq. (7). This reliance is significant as more accurate methods like Woodfisher [R11] struggle with scalability in larger models like Llama2-7B. Consequently, an imprecise estimate of $\gamma$ could degrade WAGLE’s performance, as noted in Remark 1 (Line 254).
>
> **Response to W5&Q3:** We appreciate the reviewer's feedback on the sensitivity of $\gamma$. In Remark 1 (Line 254), we detailed the examination of $\gamma$'s impact, offering guidance on its selection based on the unlearning dataset type. Our experiments (Lines 402-423) further explored $\gamma$'s sensitivity, determining its choice from insights in Remark 1. We found that a smaller $\gamma$ suits scenarios where the retain set closely mirrors the training distribution, indicated by smaller gradient norms on the retain set. Conversely, a larger $\gamma$ is advisable when there's a significant disparity between the retain and training sets, necessitating more substantial weight adjustments.
>
> **Response to Q2:** We've explored catastrophic forgetting in the context of fictitious unlearning using the TOFU dataset. Specifically, we used metrics in Table 1 to assess LLM responses on general World Facts, a task distinct from TOFU's primary focus on unlearning fictitious authors' profiles. Our results show that WAGLE not only matches or exceeds other methods in unlearning efficacy but also maintains utility in unrelated tasks. For instance, with the PO method, WAGLE improves unlearning efficacy (UE Avg.) while preserving performance on World Facts tasks (see Acc. on World Facts). This demonstrates WAGLE's ability to selectively unlearn specific information without significant catastrophic forgetting.
>
> **Response to Q4:** Our explanation is that changes in q and k require the model to relearn the attention distribution and those modules primarily capture input patterns [R12], changing on those modules will hurt utility a lot. In contrast, self-attention output (o) and value (v) modules directly influence the model's representations and outputs, making them more effective for unlearning. Additionally, editing neurons in MLP layers has also been shown to be effective for modifying model behavior [R13].
>
> **Response to Q5&Q6:** Our decision to use the 7B model size and LLaMA/Zephyr was driven by two factors: the need for direct comparison with established unlearning benchmarks like TOFU or WMDP (as referenced in [R3, R7], which used 7B model), and the computational resources available to us, particularly constrained by the GPU memory limits of our RTX A6000-equipped platform.
>
>
> **Response to Q7&Q8:**  Thank you for these suggestions. We response them as follows:
>
> - Model compression and transfer Learning: Our weight attribution framework, outlined in Eq. (2), is tailored for unlearning but also applicable to model compression and transfer learning. By adjusting the upper-level problem in our bi-level formulation, we can optimize weights for transfer learning or enhance model compression. We've linked our weight attribution scores to SNIP-based pruning [R14] in Lines 248-253.
>
> - Combining with other unlearning techniques: At submission, we were unaware of any papers combining LLM unlearning with data augmentation or adversarial training. While adversarial training [R15] is computationally intensive for LLMs, applying data transformations to forget data could enhance unlearning robustness. We consider these combinations a promising avenue for future research and will discuss these further in our future work section.
>
>
> **Response to L3.1&3.2&3.3&3.4:**  Thank you for emphasizing these key points. We will update our Limitations section accordingly:
>
> 1. Our method may introduce bias in Hessian estimation, managed by a scalar hyperparameter.
>
> 2. Approximate unlearning could compromise privacy, echoing risks associated with fine-tuning, as referenced in [R16] and [R17].
>
> 3. Adversarial robustness in LLM unlearning is under-explored. Our research, which aims to improve unlearning efficacy while maintaining utility, acknowledges vulnerabilities similar to post-fine-tuning risks [R16], identifying this as a future research area.
>
> 4. The environmental impact of unlearning is minimal in terms of additional computational demands, as discussed in responses to Q5 & Q6.
>
> **Response to L4&L5:** We cover limitations, ethical implications, etc., in Appendices E and F. In the revision, these will be integrated into the main paper for a more thorough discussion. The recommended paper will also be added to the related work section.

---

> > ### Comment · Reviewer_QvH1 · 2024-08-14
> > **Response to authors**
> >
> > I thank the authors for their detailed response. I have now a much clearer idea of my concerns and I would like to keep my score. I look forward to seeing all of the changes in the next version.

---

> > > ### Author Response · Authors · 2024-08-14
> > > **Thanks!**
> > >
> > > Dear Reviewer QvH1,
> > >
> > > Thank you very much for acknowledging our detailed response and for maintaining your positive assessment of our work. We are glad to hear that our rebuttal has helped to clarify your concerns. We are committed to incorporating your valuable feedback and our response in the revision.
> > >
> > > Thank you very much,
> > >
> > > Authors

---

### Official Review · Reviewer_MRmy · 2024-07-11

**Soundness:** 3
**Presentation:** 2
**Contribution:** 2
**Rating:** 4
**Confidence:** 3

**Summary:**

This paper presents WAGLE, a weight attribution method for unlearning. The main
goal of WAGLE is to augment existing unlearning methods by identifying
influential parameters, and restricting existing methods to the
influential parameters. The authors focus their attention to approximate unlearning for large language models (LLMs).

To find influential parameters, the authors propose to model the approximate unlearning problem as a bi-level optimization (BLO) problem, and then use classic tools from the BLO literature to derive an efficient solution.

The authors evaluate WAGLE on three different unlearning methods. As baselines, the authors consider three pruning techinques together with  LoRA, as well as no pruning.

**Strengths:**

The authors provide an extensive and thorough evaluation of their proposed method. In particular, they consider 1) multiple base unlearning algorithms, 2) multiple well-motivated pruning baselines, 3) multiple unlearning efficacy heuristics, and 4) multiple downstream tasks to evaluate model utility.

Another strength on the experimental side is that experiments are performed on Llama2-7B; the size of the model alone displays the scalability of the proposed method.

**Weaknesses:**

- Motivation: "Yet, effective unlearning also requires a sense of locality, which involves
identifying the sub-components of the LLM (i.e., a subset of weights in this
work) that are crucial for the unlearning task, while minimally impacting the
model’s original utility." (lines 167-169)

I am not sure I'm convinced by this assertion, especially given that it is provided without any supporting evidence. In particular, the null hypothesis for me here would be that unlearning a given subset of the training data would require modifying most (if not all) model parameters. A simple example can be observed in linear regression, where we have a closed form for model updates via the Woodbury-Sherman-Morrison formula. Even in that simple scenario, there is no notion of sparsity in the unlearning update. Given that this statement is central to the entire paper, I would like to see an argument backing it up.

- Results are improved only for one UE metric (FQ), even when comparing against the trivial "random pruning" baseline. In particular, on all three unlearning methods in Table 1, the performance of "random" and WAGLE on MIA,  1-FA, 1-Rouge-L seems within statistical error (see Q below about providing SEs).

**Questions:**

- Figure 1: " Yet, when applied to LLM unlearning, the effectiveness of weight sparsity remains elusive" (lines 179-180)
From Figure 1, it seems like at 0% sparsity, the model is completely useless (0% model utility), and at 5% sparsity (obtained with Wanda pruning),  there is some balance between model utility and unlearning efficacy. What happens at more granular levels of sparsity between 0% and 5%? Does efficacy degrade gracefully, allowing for a satisfactory trade-off between utility and efficacy?

To me, this question is important, because the existence of a "good" sparsity level here would mean that the unlearning landscape for LLMs follows that for classification models, where, as the authors note, sparsity has been shown to be helpful.  Thus, such a finding would make weight attribution methods unnecessary for this application, given their additional complexity and computational overhead.

- Can the authors provide wall-clock time estimates for WAGLE and the baselines they consider?
- Can the authors provide standard errors in Table 1?
- Is there a connection with influence functions

---

> ### Author Rebuttal · Authors · 2024-08-07
>
> Thank you for your insightful feedback. Below, we provide detailed responses to each weakness (W) or question (Q). References (in the format of [Rx]) can be found in the general response.
>
>
>
> **Response to W1:**  Thank you for your comments on the importance of locality in effective unlearning. Here are additional clarifications:
>
> First, the concept of "effective unlearning requires a sense of locality" has been empirically supported by existing methods discussed in Lines 60-69. In particular, in the context of GA-based unlearning for discriminative classifiers, it has been theoretically shown that imposing model sparsity can significantly reduce the performance gap between approximate unlearning (which is computationally efficient but not verified) and exact unlearning (i.e., retraining from scratch) [R1]. This prior evidence underscores the importance of locality in enhancing the efficacy of approximate unlearning.
>
> Second, the linear regression example proposed by the reviewer is insightful. As we explained above, locality is suitable for improving approximate unlearning. Yet, it may not be necessary for exact unlearning if you can readily achieve. This aligns with the reviewer’s argument that a closed form for model updates via the Woodbury-Sherman-Morrison formula is sufficient for unlearning in linear regression.
>
> Third, we would also like to refer the reviewer to the **GR1** for further clarification on the non-trivialness of finding proper locality by weight attribution and the associated benefit in improving the unlearning efficacy-utility preservation tradeoff for existing unlearning methods.
>
> In the revision, we will make the above points clearer.
>
> **Response to W2&Q3:** Thank you for your inquiry regarding our evaluation metrics and the statistical significance of our results.
>
> First, the best assessment of unlearning efficacy (UE) presents challenges in the literature, prompting us to utilize various UE metrics to ensure a comprehensive evaluation. It is important to note that different unlearning methods demonstrate different sensitivities to these metrics. For instance, we agree with the reviewer that the FQ metric is more distinguishable, particularly under the GradDiff and PO methods detailed in Table 1. For the NPO method, however, metrics like 1-FA and 1-Rouge-L are more responsive, where our method also consistently outperforms other baselines.
>
> Additionally, the compatibility between unlearning objectives and attribution methods varies significantly. For instance, the Wanda pruning method exhibits substantially lower unlearning efficacy (UE) across all metrics when using gradient ascent-type unlearning objectives in GA and its extended version, NPO. In contrast, our method demonstrates robust UE performance regardless of the unlearning objectives applied. To facilitate easier comparison and provide a general ranking of the different methods, we calculated an average of these UE metrics (UE Avg.), where WAGLE consistently emerges as the top performer.
>
> Lastly, to address questions about statistical significance, we've added standard errors in **Table R1** of the attached PDF, confirming that WAGLE's improvements over the 'random' baseline are significant. For instance, WAGLE improves UE Avg. by 0.03 with a standard error of 0.006, and maintains or enhances UT Avg. The GradDiff method shows a 0.048 improvement in UE Avg. with a standard error of 0.01, and NPO method increases utility by 0.1715 with standard error of 0.01, while maintaining similar unlearning efficacy as the random baseline. Additionally, under multiple runs, the random baseline exhibits a larger variance overall. For instance, 'random' shows a larger variance in UE Avg. (0.0286) compared to our method (0.0105) on GradDiff method, and a larger variance in UT Avg. (0.0124) compared to our method (0.0012).
>
> We will clarify the above points in the revision.
>
> **Response to Q1:**  In response to your query about the effects of finer levels of sparsity between 0% and 5%, we conducted additional experiments using Wanda pruning across these granular levels on the TOFU dataset, as depicted in Figure 1. The detailed results are shown in **Table R3** of the attached PDF.
>
> The finer-level results revealed that the interaction between Wanda pruning and the NPO unlearning method exhibits a particularly sensitive trade-off between unlearning efficacy (UE Ave.) and utility (UT Avg.). For instance, at a 1% sparsity level with Wanda + NPO, the utility scores slightly higher at 0.6140 compared to 0.5936 for WAGLE + NPO. However, the unlearning efficacy for Wanda + NPO drops significantly to 0.8377, contrasting sharply with 0.9641 for WAGLE + NPO in Table 1. This underscores that while conventional LLM pruning might marginally improve utility, it does not enhance unlearning efficacy, reflecting its poor suitability for unlearning tasks where weight attribution should balance unlearning ‘objective’ with utility ‘constraint’.
>
> **Response to Q2:**  Please refer to **GR2** in the general response.
>
> **Response to Q4:** Thank you for your insightful question. Indeed, there is a connection between influence functions and our approach. In the context of unlearning, influence functions are commonly used to assess the impact of specific data points on a model, a process known as influence unlearning [R1,R8]. However, the utility of influence functions extends beyond mere data influence analysis. Influence functions are also powerful tools for solving bi-level optimization problems [R9], where the solution to a lower-level problem influences the outcomes at an upper level, such as evaluating model weight influence in Eq. (2). In our approach, we employ an influence function to compute the implicit gradient necessary for weight attribution, which we frame as a bi-level optimization problem. This method allows us to systematically assess and manage the impact of specific weights on unlearning efficacy and utility preservation.

---

> > ### Comment · Reviewer_MRmy · 2024-08-13
> >
> > I appreciate the response by the authors. However, my concerns regarding evaluation remain and thus I am keeping my rating.

---

> ### Author Response · Authors · 2024-08-13
> **Thank you and follow-up inquiry**
>
> Dear Reviewer MRmy,
>
> Thank you for acknowledging our response. We regret that some of your concerns regarding our evaluation persist. Could you please provide more specific details about the unresolved aspects? As you can see, we have dedicated significant effort to address your initial comments through both our general and individual responses. If there are additional clarifications or information you require, we are more than willing to provide them before the end of the discussion phase.
>
> Thank you once again for your engagement and feedback.
>
> Authors,

---

### Official Review · Reviewer_eHyY · 2024-07-13

**Soundness:** 4
**Presentation:** 3
**Contribution:** 4
**Rating:** 8
**Confidence:** 4

**Summary:**

This paper explores the correlation between model weights and the efficacy of unlearning processes in large language models (LLMs). It introduces a framework called WAGLE (Weight Attribution-Guided LLM Unlearning), which elucidates the relationship between the influence of specific weights and the LLM unlearning performance. WAGLE is grounded in a meticulous analysis from the influence function perspective. Moreover, the paper evaluates the framework's performance across a broad set of benchmarks and applications specific to LLM unlearning, demonstrating its superior effectiveness over traditional dense unlearning methods.

**Strengths:**

1.	This paper is pioneering in the domain of machine unlearning for large language models (LLMs), providing a formal definition and derivation that demonstrates how weight attribution contributes to unlearning processes. Unlike prior studies on unlearning in computer vision, which were primarily intuitive, this paper’s methodical approach through formal derivation introduces a significant novelty to the field.

2.	The insights offered by the proposed framework are substantial, particularly in identifying critical modules within LLMs that are pivotal for effective unlearning, as detailed in Figure 2. This not only deepens our understanding of the inner workings of LLMs but also highlights specific areas for targeted unlearning strategies. This may also helps understanding the memory of LLMs.

3.	The clarity and organization of the paper enhance the reader's comprehension and engagement.

4.	It features a comprehensive set of experiments across diverse tasks, benchmark datasets, and models, convincingly demonstrating the efficacy of the proposed weight attribution methodology in facilitating LLM unlearning.

**Weaknesses:**

1.	My primary concern lies with the clarity of the evaluation metric, Membership Inference Attack (MIA), as utilized on the TOFU dataset. The paper does not clearly define whether a lower MIA score, which would indicate a reduced likelihood of the forget set being recognized as part of the training set, is preferable. Clarification on this metric and its desired outcome would enhance the understanding of the results presented.

2.	The experiments primarily focus on a model size of approximately 7 billion parameters. It would be beneficial if the authors could explore how the unlearning performance varies with changes in model size. This would provide insights into the scalability and applicability of the proposed method across different model architectures.

3.	Another concern is the computational efficiency of the unlearning process, specifically the time cost associated with implementing weight attribution. Providing comparative data on the time required for unlearning with and without weight attribution would significantly strengthen the paper by highlighting the practical implications of the proposed method.

**Questions:**

1.	Could the authors provide additional clarification regarding the implementation and interpretation of the Membership Inference Attack (MIA) evaluation metric on the TOFU dataset?

2.	Could the authors explore how the efficacy of the weight attribution method in facilitating LLM unlearning varies across models of different sizes?

3.	Could the authors provide details on the time cost associated with their unlearning method, both with and without the implementation of weight attribution?

**Limitations:**

The limitation concerning the selection of an appropriate sparsity level for weight attribution is a recognized challenge within the broader pruning community and remains an unresolved question.

---

> ### Author Rebuttal · Authors · 2024-08-07
>
> Thank you for your thorough review, positive assessment, and insightful feedback on our submission. Below, we provide detailed responses to each identified weakness (W) and question (Q). References (in the format of [Rx]) can be found in the general response.
>
> **W1/Q1: Could the authors provide additional clarification regarding the implementation and interpretation of the Membership Inference Attack (MIA) evaluation metric on the TOFU dataset?**
>
> **A:** Thank you for requesting further clarification on the implementation and interpretation of the Membership Inference Attack (MIA) evaluation metric on the TOFU dataset. We are sorry for any confusion if our initial explanation lacked clarity.
>
> We employ the Min-k% Probability method [R5] as a "predictor" in the post-unlearning phase to assess whether test-time forget data were part of the LLM's original training set (Lines 309-312). Post-unlearning, the ideal case is that forgotten data should not be recognized as part of the training dataset, akin to test set data. In this context, correctly identifying forgotten samples as non-training instances equates to "true negatives" in our MIA approach (Appendix C.3, [R1]).
>
> Based on the above, we measure the effectiveness of unlearning by calculating the AUROC for this MIA detector, rather than merely tallying true negatives. This involves considering both the forget set (treated as non-training data) and the retain set (data that remains part of the training set). A higher AUC indicates that the model accurately distinguishes between forgotten and retained data, reflecting more effective unlearning. This shows the MIA’s capability when evaluating unlearned LLMs to correctly classify forget and retain data points as belonging outside or inside the training set, respectively, thereby demonstrating successful unlearning.
>
>
> **W2/Q2: It would be beneficial if the authors could explore how the unlearning performance varies with changes in model size.**
>
> **A:**  We appreciate your emphasis on the importance of scaling in LLMs. Here, we make the following clarifications.
>
> (Rationale behind our choice) First, our choice of using the 7B model sizes is consistent with established benchmarks (e.g., TOFU) in the literature on LLM unlearning [R3, R6, R7]. Focusing on this specific model  facilitates us to compare the unlearning performance of our proposal with other baselines. Second, our choice of model sizes was influenced by the GPU resources available to us. Our primary computational platform, equipped with RTX A6000 GPUs, imposes memory limitations on the feasible model size of LLMs we can experiment with extensively. This constraint affected our ability to perform extensive experiments on larger models, such as those exceeding 10B parameters. We will clarify this limitation in the section Limitations (Appendix E).
>
>
> **W3/Q3: Another concern is the computational efficiency of the unlearning process, specifically the time cost associated with implementing weight attribution. Providing comparative data on the time required for unlearning with and without weight attribution would significantly strengthen the paper by highlighting the practical implications of the proposed method.**
>
> **A:** Please refer to **GR2** in the general response.

---

> > ### Comment · Reviewer_eHyY · 2024-08-13
> > **update**
> >
> > Thank you for the detailed responses and additional experiments provided during the rebuttal phase. I appreciate your efforts to address my initial concerns. As a result, I have raised my evaluation score to 8.

---

> > > ### Author Response · Authors · 2024-08-13
> > > **Thank you!**
> > >
> > > Dear Reviewer eHyY,
> > >
> > > Thank you for acknowledging the efforts we put forth during the rebuttal process. We are heartened to learn that our responses have addressed your initial concerns satisfactorily, and we are grateful for your decision to raise the score to 8. We will ensure to improve our paper in the revision, incorporating your valuable insights to enhance its quality.
> > >
> > > Thank you once again for your thorough review and for recognizing the adequacy of our responses.
> > >
> > > Best regards,
> > >
> > > Authors

---

### Official Review · Reviewer_BmWx · 2024-07-14

**Soundness:** 3
**Presentation:** 3
**Contribution:** 3
**Rating:** 7
**Confidence:** 4

**Summary:**

This paper introduces WAGLE (Weight Attribution-Guided LLM Unlearning), a novel method for large language model (LLM) unlearning that identifies influential weights for the unlearning process while considering the retain loss. The authors evaluate WAGLE on a diverse set of unlearning benchmarks, demonstrating improved forgetting performance and competitive task performance compared to baseline methods.

**Strengths:**

* Novel approach: WAGLE integrates weight analysis with unlearning, providing a new perspective on LLM unlearning.
* Comprehensive evaluation: The method is tested on various unlearning tasks and datasets, showcasing its versatility.
* Improved performance: WAGLE consistently enhances forgetting ability while maintaining or sometimes improving task performance on retain sets.
* Insights into model architecture: The study reveals that early-to-mid layers are important for unlearning, and the method tends to avoid selecting weights from self-attention mechanisms.
* Comparison with existing methods: The authors benchmark WAGLE against various direct optimization approaches for forget/retain loss, providing context for its performance.
* The bi-level optimization formulation is interesting and could be built on by future methods.

**Weaknesses:**

* Discussion of limitations and future is deferred to the appendix, it would be good to include this in the main paper
* It isn't clearly articulated *why* this method improves existing unlearning methods. Is it faster or easier to train? Is it hard to get improvements on forgetting performance?
* Concerns about evaluation: It seems like the evaluation is not quite appropriate, especially if it is hard to get improvement in forgetting performance. It is not clear to me how the tradeoffs between the retain and forget loss are set. Given the nature of the evaluation, it seems important to at least discuss how this was tuned (apologies if I missed it). In an ideal world, the methods would be evaluated across a sweep of this variable. It would be very nice to know how much retain loss the comparison methods need to give up in order to match the forget loss performance of WAGLE.

**Questions:**

Please respond to the concerns raised in the weaknesses section to provide additional context for your choices or suggest ways to address the issue. Thank you.

**Limitations:**

There is some discussion of this in the appendix, but it should be included in the main text. A bit more discussion of the societal ramifications of unlearning (potential benefits or risks) would also be nice.

---

> ### Author Rebuttal · Authors · 2024-08-07
>
> Thank you for your insightful comments. Below, we address each identified weakness (W) in detail. References (in the format of [Rx]) can be found in the general response.
>
> **W1: Discuss limitations and future work in the main paper rather than the appendix.**
>
> **A:** We will follow your suggestion to move the discussion of limitations and future work from the appendix to the main paper in the revision.
>
> **W2: Why does this method improve on existing unlearning methods?**
>
> **A:** Please refer to **GR1** in the general response.
>
> **W3: Concerns about evaluation: It seems like the evaluation is not quite appropriate, especially if it is hard to get improvement in forgetting performance. It is not clear to me how the tradeoffs between the retain and forget loss are set. Given the nature of the evaluation, it seems important to at least discuss how this was tuned (apologies if I missed it). In an ideal world, the methods would be evaluated across a sweep of this variable. It would be very nice to know how much retain loss the comparison methods need to give up in order to match the forget loss performance of WAGLE.**
>
> **A:** Thank you for sharing this concern. We would like to make the following clarifications.
>
> First, regarding our evaluation (e.g., in  Table 1), we included multiple metrics to provide a comprehensive assessment of unlearning performance. These metrics do not always follow the same trend. For instance, a collapsed model might perform well on 1-FA and 1-ROUGEL for measuring unlearning efficacy but poorly on metrics like MIA and Forget Quality (FQ). This is because MIA and FQ measure differences between outputs from unlearned models on forget sets and retain sets, where a collapsed model could treat both sets the same, leading to lower performance on these metrics. Thus, for ease of comparison, we normalized the metrics to a [0,1] range and used their average values to give a general ranking. **Figure R1** in the attached PDF showed a clear trend of improved tradeoff using WAGLE.
>
> Second, we have added results over three independent trials and reported the standard deviations in **Table R1.** This table demonstrates that our method achieves better unlearning efficacy compared to baselines, as indicated by higher UE Avg. in bold. The improvement is statistically significant relative to the standard deviation. Additionally, the utility of our method (as indicated by UT Avg.) matches or even surpasses that achieved by unlearning using dense models.
>
>
> Third, regarding the tuning of the trade-offs between retain and forget loss, specifically the hyperparameter $\lambda$, we followed the standard TOFU benchmark settings as outlined in references [R3,R4], which consistently set $\lambda$ to 1. This is discussed in Appendix B.4. To provide a clearer understanding, we have conducted additional experiments during the rebuttal phase, as shown in **Table R2**. We varied $\lambda$ across a range of values (between 0.1 and 10) using different unlearning methods for the dense LLM (that excludes weight attribution). The results indicate that UE Avg. consistently decreases while UT Avg. increases as $\lambda$ increases across different unlearning methods. This is not surprising as a larger $\lambda$ indicates a higher penalty on minimizing the utility loss on the retain set. **Table R2** shows that even with tuning, dense models do not achieve as favorable a trade-off between utility and unlearning efficacy as WAGLE with the default $\lambda$ setting.
>
> This additional analysis underscores that WAGLE's importance  is not merely due to a specific $\lambda$ setting but rather due to its inherent weight attribution ability to achieve unlearning while taking utility optimization into account, i.e., balancing unlearning 'objective' with utility 'constraint'.
>
>
>
> **W4: There is some (limitations) discussion of this in the appendix, but it should be included in the main text. A bit more discussion of the societal ramifications of unlearning (potential benefits or risks) would also be nice.**
>
> **A:** Following your suggestion, we will revise the paper by discussing limitations in the main paper and add more discussions on societal ramifications.

---

### Author Rebuttal · Authors · 2024-08-07

We appreciate the detailed feedback from all reviewers. Below is a general response addressing common concerns highlighted in your comments. Refer to the attached PDF for figures and tables labeled as **Figure Rx** and **Table Rx**, where 'R' denotes 'rebuttal'.



**GR1: Why weight attribution helps LLM unlearning? (@Reviewers BmWx, MRmy, QvH1)**

**A:** To better clarify why weight attribution helps LLM unlearning, we provide this general response.

Our method, WAGLE, is not aimed at speeding up the training process for unlearning but rather at enhancing the balance between unlearning effectiveness and utility retention in LLM unlearning.

As outlined in Lines 62-64 and 176-178, prior research [R1] demonstrates that incorporating "proper" sparsity can reduce errors in approximate unlearning relative to exact unlearning. However, localizing the crucial subset of weights for LLM unlearning is challenging, as indicated in the motivation section starting from Line 165: Existing LLM pruning methods [R2] show inadequate for this task. As shown in Fig. 1, utility suffers a significant drop when unlearning stays effective against model sparsity. That is, the optimal tradeoff between unlearning effectiveness and utility retention is highly non-trivial to achieve using conventional LLM pruning.    WAGLE addresses this by exploring and exploiting weight attribution, seeking the optimal  trade-off between unlearning efficacy and model utility as reflected in our bi-level problem (2) in the section "Balancing unlearning 'objective' with utility 'constraint'. This approach leads to better unlearning without compromising utility or vice versa.

To substantiate our rationale, **Figure R1** in the attached PDF depicts the trade-off between unlearning effectiveness and model utility across different methods on the TOFU dataset, where the legend label 'WAGLE' and 'Dense' indicate the application of weight attribution or not. The x-axis shows average utility (UT Avg.), and the y-axis shows average unlearning efficacy (UE Avg.); See the dense and WAGLE rows in Table 1 as well. For both metrics, higher values indicate better performance. Ideally, the best LLM unlearning method would appear in the upper right corner. As we can see from **Figure R1**, all unlearning objectives (GradDiff, PO, and NPO) benefit from WAGLE, enhancing one performance aspect without compromising the other, unlike traditional unlearning under dense models.

**GR2: Computational Cost Comparison between WAGLE and Baselines (@ Reviewers eHyY, MRmy, QvH1)**

**A:**  Thank you for raising this question. First, as indicated by Eqs. (7)-(8), the weight attribution mask can be computed offline using only first-order derivatives. As a result, generating a general unlearning mask for the TOFU dataset takes approximately 4 minutes on the Llama2-7B-chat model, as shown in **Table R4**. Second, applying the mask during the unlearning process requires similar running time across different unlearning methods, as shown in **Table R4**. Considering the overall 30-minute unlearning process, the time required to generate the attribution mask is relatively minimal.


**GR3: A summary of additional experiments (@All reviewers).**



**A:** We have made a substantial effort to enrich our experiments based on reviewers’ suggestions (see the attached PDF). Below is a summary, where Q-i (or W-i) represents the $i$-th question (or weakness) in our individual responses:

**Reviewer BmWx:**

- W3: Ablation study on effect of $\lambda$ for TOFU task (**Table R2**)
- W2: Clearer visualization for WAGLE advantage on improved tradeoff between unlearning effectiveness and utility preservation  compared with baselines (**Figure R1**)

**Reviewer eHyY:**

- W2/Q2: Time comparison between WAGLE and baseline methods (**Table R4**)

Reviewer MRmy:

- W2 & Q3: Multiple runs to adding standard error to Table 1. (**Table R1**)
- Q1 : Unlearning performance of finer levels of sparsity between 0% and 5% on the TOFU dataset (**Table R3**)
- Q2 : Time comparison between WAGLE and baseline methods (**Table R4**)

**Reviewer QvH1:**
- W2 & Q1: Time comparison between WAGLE and baseline methods (**Table R4**)

**References used in authors' response:**
> [R1] Jia, et al. "Model sparsity can simplify machine unlearning." NeurIPS,2023.
>
> [R2] Sun, et al. "A simple and effective pruning approach for large language models." arXiv, 2023.
>
> [R3] Maini, et al. "Tofu: A task of fictitious unlearning for llms." arXiv, 2024.
>
> [R4] Zhang, et al. "Negative preference optimization: From catastrophic ..." arXiv, 2024.
>
> [R5] Shi, et al. "Detecting pretraining data from large language models." arXiv, 2023.
>
> [R6] Yao, et al. "Large language model unlearning." arXiv, 2023.
>
> [R7] Li, et al. "The wmdp benchmark: Measuring and reducing malicious use with unlearning." arxiv, 2024
>
> [R8] Pang, et al. Understanding black-box predictions via influence functions.ICML, 2017.
>
> [R9] Zhang, et al. "An Introduction to Bilevel Optimization: Foundations and ...." IEEE Signal Process. Mag.,2024.
>
> [R10] Shi, et al. "MUSE: Machine Unlearning Six-Way Evaluation for Language Models." arXiv, 2024.
>
> [R11] Singh, et al. "Woodfisher: Efficient second-order approximation for neural network compression." NeurIPS,2020.
>
> [R12] Geva, et al. "Transformer feed-forward layers are key-value memories." arXiv, 2020.
>
> [R13] Meng, et al. "Locating and editing factual associations in GPT." NeurIPS, 2022.
>
> [R14] Lee, Namhoon, Thalaiyasingam Ajanthan, and Philip HS Torr. "Snip: Single-shot network pruning based on connection sensitivity." arXiv, 2018.
>
> [R15] Madry, et al. "Towards deep learning models resistant to adversarial attacks." arXiv, 2017.
>
> [R16] Qi, et al. "Fine-tuning aligned language models compromises safety, even when users do not intend to!." arXiv, 2023.
>
> [R17] Hayes, et al. "Inexact unlearning needs more careful evaluations to avoid a false sense of privacy." arXiv, 2024

---

### Decision · Program_Chairs · 2024-09-25

**Decision:**

Accept (poster)

**Comment:**

This paper addresses the problem of unlearning, suggesting a method for identifying "influential weights". The paper received an engaging discussion and following rebuttal stands with scores of 8,7,4,7 with the 4 participating relatively little in the discussion.

The authors make a claim "Yet, effective unlearning also requires a sense of locality, which involves identifying the sub-components of the LLM" which reviewer MRmy is rightly concerned about. This appears to me to be at-best a speculative assertion that cannot be made by fiat and I expect the authors to soften this claim to an expression of intuition "we imagine that one promising path towards progress on unlearning might be to identify local sub-components of LLMs ...".

Otherwise, reviewers appreciated the novelty of the approach, the clarity and organization of the paper, the comprehensiveness of the evaluation, insights regarding model architecture, and the degree of comparison to existing methods.

Overall, given the broad support for the paper, I am recommending acceptance.